# AV-Cloud: Spatial Audio Rendering Through Audio-Visual Cloud Splatting

**Mingfei Chen** *                              **Eli Shlizerman** [†‡]

## Abstract

We propose a novel approach for rendering high-quality spatial audio for 3D scenes that is in synchrony with the visual stream but does not rely or explicitly conditioned on the visual rendering. We demonstrate that such an approach enables the experience of immersive virtual tourism - performing a real-time dynamic navigation within the scene, experiencing both audio and visual content. Current audio-visual rendering approaches typically rely on visual cues, such as images, and thus visual artifacts could cause inconsistency in the audio quality. Furthermore, when such approaches are incorporated with visual rendering, audio generation at each viewpoint occurs after the rendering of the image of the viewpoint and thus could lead to audio lag that affects the integration of audio and visual streams. Our proposed approach, *AV-Cloud*, overcomes these challenges by learning the representation of the audio-visual scene based on a set of sparse AV anchor points, that constitute the Audio-Visual Cloud, and are derived from the camera calibration. The Audio-Visual Cloud serves as an audio-visual representation from which the generation of spatial audio for arbitrary listener location can be generated. In particular, we propose a novel module Audio-Visual Cloud Splatting which decodes AV anchor points into a spatial audio transfer function for the arbitrary viewpoint of the target listener. This function, applied through the Spatial Audio Render Head module, transforms monaural input into viewpoint-specific spatial audio. As a result, AV-Cloud efficiently renders the spatial audio aligned with any visual viewpoint and eliminates the need for pre-rendered images. We show that AV-Cloud surpasses current state-of-the-art accuracy on audio reconstruction, perceptive quality, and acoustic effects on two real-world datasets. AV-Cloud also outperforms previous methods when tested on scenes "in the wild".

## 1   Introduction

With the advent of visual rendering for 3D scenes, it is now possible to reconstruct a scene as a set of cloud points from a collection of real-world images or videos [1, 2, 3]. It is then possible to embark on virtual tourism through the scene, which would traverse it through novel paths and various viewpoints in real-time, where in each frame, the scene is rendered with remarkable photo realism. While visual realism and visual perception of current virtual tourism methods are striking [3, 4, 5, 6, 7], additional modalities such as sound could enhance and contribute to a fuller, more dynamic, and more immersive experience. Imagine attending a virtual tour of the *Trevi Fountain* in Rome, where you can not only view the fountain from different locations but also hear its magical sounds. Furthermore, these sounds would adapt based on your view perspectives while exploring the scene.

Indeed, as visual rendering creates 3D scenes, audio rendering produces dynamic spatial sounds, adding the natural auditory perception to the perception of the scene. Combined together, visual and

---

*Department of Electrical & Computer Engineering, University of Washington, Seattle, USA

[†]Department of Applied Mathematics, University of Washington, Seattle, USA

[‡]Corresponding author: shlizee@uw.edu

38th Conference on Neural Information Processing Systems (NeurIPS 2024).

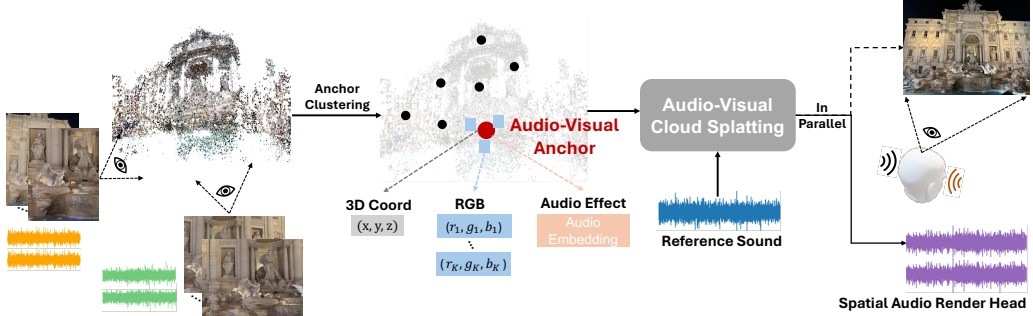

Figure 1: AV-Cloud is an audio rendering framework synchronic with the visual perspective. Given video collections, it constructs Audio-Visual Anchors for scene representation and transforms monaural reference sound into spatial audio.

audio components provide a cohesive, immersive experience that allows to explore and interact within virtual spaces. The creation of such an experience requires the rendering of *high-quality spatial audio* which is *synchronized with the visual rendering*. In addition to real-time speed requirements, this task is challenging since such audio rendering requires careful modeling of comprehensive factors such as magnitude changes and spatial effects (e.g., left-right channel energy ratios for stereo audio) that match with view perspectives and reverberation that enhances the sound's situational presence within a scene. While, in principle, the spatial audio can be rendered in a scene by comprehensive modeling of the Room Impulse Response (RIR) [8, 9, 10, 11, 12, 13, 14, 15, 16, 17, 18, 19, 20, 21] that will be convolved with sounds emitted from emitters, achieving high fidelity modeling of RIR is reserved for synthetic scenes with simulated RIR or scenes that have been thoroughly scanned for such purpose. Indeed, accurate modeling of RIR demands a thorough understanding of the scene properties such as the scene geometry and surface materials, all of which are challenging to estimate for scenes reconstructed from "in-the-wild" sparse collections of images or videos. Furthermore, knowledge of the locations of the emitters will need to be known to reconstruct the spatial sound.

To address the challenges of RIR reconstruction for various real scenes, alternative methods propose to reconstruct spatial audio based on images [22, 23, 24, 25, 26, 27, 28, 29] at the target viewpoint. Such an approach could be advantageous since could apply to in-the-wild reconstructed scenes with an easier setup and set a suitable audio renderer which follows visual rendering. While such image-based approaches pave the way to audio-visual rendering, their accuracy relies on the visual accuracy of rendered images such that visual artifacts will impact the sound quality. Additionally, by design, such audio rendering is sequential and occurs after visual rendering. These factors create a lag that needs to be constantly addressed to achieve synchrony in audio-visual rendering, affecting real-time audio-visual rendering by preventing the seamless integration of audio and visual streams.

In this work, we propose to address the above described challenges by introducing a novel framework, *AV-Cloud*, which is a point-based audio-visual rendering framework that supports high-quality spatial audio rendering from any viewpoint. AV-Cloud does not rely on pre-rendered visual cues (e.g. rendered images) and thus enables spatial audio rendering simultaneously with visual rendering.

Specifically, as illustrated in Figure 1, AV-Cloud approach starts from a sparse set of Structure-from-Motion (SfM) points [1], derived from camera calibration and estimated from a collection of real-world videos. SfM points provide detailed 3D scene geometry for accurate spatial rendering and serving as consistent initial inputs for synchronizing visual and audio rendering processes. AV-Cloud then establishes clusters of these points and initializes representative Audio-Visual Anchors (AV anchors), each bound with 3D coordinates, RGB visual features, and audio effect latent features for a point-based audio-visual scene representation. AV anchors then undergo splatting by the Audio-Visual Cloud Splatting (AVCS) module, which decodes them into spatial audio transfer function by dynamically adjusting the contribution of each anchor based on the viewpoint pose of the listener. Together with the Spatial Audio Render Head (SARH), AV-Cloud converts monaural reference sound into viewpoint-specific stereo audio, effectively adjusting the acoustic effects. Optimization of the reusable AV-Cloud across all training samples allows for effective generalization of spatial audio rendering to accommodate novel viewpoints while using a minimal set of parameters.

In summary, our main contributions in this work are as follows: 1) We introduce a novel point-based audio-visual rendering framework, AV-Cloud, which enables high-quality audio rendering from any

viewpoint without reliance on pre-rendered cues and thus better preserves synchrony with visual rendering. 2) We define Audio-Visual Anchors for scene representation and propose the Audio-Visual Cloud Splatting module. This module, along with a Spatial Audio Render Head, dynamically adjusts AV anchor contributions based on view perspectives to render high-quality stereo sound with accurate acoustic effects. 3) Our framework surpasses state-of-the-art(SOTA) accuracy in spatial audio reconstruction, perceptual quality, and acoustic effects on two common real-world benchmarks. It also outperforms existing methods in experiments "in the wild" at a real-time inference speed.

## 2 Related Works

**Spatial Audio Scene Reconstruction.** Most traditional audio scene reconstruction methods reconstruct spatial sound at arbitrary listener locations by convolving the sound waveform of each emitter with corresponding Room Impulse Response (RIR) and then perform summation of the outcomes of each emitter. Conventional RIR modeling methods were based on solving the acoustic wave equation [8, 9, 10, 11] or treat sound propagation as optic rays [12, 13, 14, 15, 16, 17]. Recent methods utilize deep learning approaches to generate spatial RIRs for emitter-receiver pairs. Particularly, in [18, 19, 20, 21], a deep generative model is learned, while alternative methods [30, 31, 32] learn an implicit neural function to represent RIR. Since RIR is the complex outcome of sound propagation through the scene geometry and interaction with surfaces, the aforementioned RIR generators are conditioned on the locations of the emitters and scene properties such as geometry [31, 32, 19], mesh [20] and visual at target viewpoint [21].

In real-world applications, the explicit emitter location and detailed scene geometry information are challenging to capture without special setups. To overcome this requirement, it was proposed to reconstruct spatial audio scenes from images, giving rise to methods such as visual acoustic matching [22, 23, 27] and visual-guided audio spatialization [24, 25, 26]. These methods primarily focused on matching environmental levels, that could introduce inaccuracies in learning acoustic effect difference during continuous viewpoint changes. To address these limitations, the study in [28] introduced a novel-view acoustic synthesis task and a neural rendering approach that learns to synthesize the sound of an arbitrary perspective in the space guided by visual images. This was followed by the study in [29] which utilized NeRF-based rendering method [33] to synthesize novel videos with spatial sound from arbitrary camera poses. Both [28] and [29] rely on viewpoint information rather than uniform scene representations for audio rendering which may result in inaccuracies when tested on novel camera poses. Furthermore, the requirement of view images as input, could reduce the audio rendering quality due to visual artifacts and introduce rendering delays. Our method renders spatial audio in synchrony with the visual rendering for any viewpoint since audio is rendered directly from the learned scene representation.

**Point-based Neural Rendering for Audio-Visual Scenes.** Point-based methods were shown to render disconnected and unstructured geometry samples such as point clouds in an efficient way [34]. Points capture the underlying data of a 3D scene, and hold essential physical information critical for precise depiction in both visual and audio rendering. In some instances, point sample rendering for visual content could lead to holes due to the extreme discontinuity of points. Traditionally, this issue was addressed by splatting the points to extents larger than a pixel [35, 36, 37, 38]. This approach helps in mitigating discontinuities. Notably, a recent method 3D-GS[3] utilizes 3D Gaussians for a flexible and expressive scene representation, leveraging explicit representation and differential point-based splatting methods for real-time rendering of novel views. In contrast to traditional point-based methods that require Multi-View Stereo (MVS) data [39, 40, 41], 3D-GS achieves high-quality results from Structure-from-Motion (SfM) points only [1] from camera calibration as input, optimizing properties such as position, opacity, anisotropic covariance at each point.

For spatial audio rendering, the mapping of 3D points to audio signals is more complex than projecting points from 3D to 2D for visual rendering. Therefore, implicit neural field representations are popular for point-based neural audio rendering [32, 42, 43, 44]. Specifically, in [42], distinct local feature grids were used for the emitter and the receiver as an inductive bias to generalize to novel inputs, while in [32], scene geometry features were disentangled with three modules to generate independent features for the emitter, the boundary points, and the listener, respectively, enhancing feature reuse. These methods rely on knowledge of the locations of the emitter and the listener, and they do not adapt to the orientation of the viewer. Such property is key for realistic spatial effects. In contrast, AV-Cloud dynamically weighs the points based on target perspectives to support view-based audio rendering.

# 3 Methods

Given a collection of videos, we aim to render spatial audio synchronized with arbitrary listener viewpoint given monaural reference sound. As shown in Figure 1, for each scene, SfM points are estimated from training videos and camera calibration [1]. We define $N$ Audio-Visual Anchors after clustering the SfM points, initializing their features on 3D coordinates, RGB and audio effect to construct point-based audio-visual representation for the 3D scene. We then introduce the Audio-Visual Cloud Splatting (AVCS) module to decode spatial audio transfer function conditioned on the listener viewpoint. The transfer function together with the Spatial Audio Render Head (SARH) will transfer the input reference sound to target stereo audio at that viewpoint. Our method renders spatial audio directly from Audio-Visual Anchors without the need for pre-rendered images or specified locations of the emitters.

## 3.1 Structure from Motion

Structure from Motion (SfM) is a technique that is used to reconstruct a three-dimensional structure of the environment from a sequence of images [1]. SfM points represent distinct, recognizable features in the scene, which are detected and used to estimate the viewpoint of the camera. This process results in a detailed 3D point cloud that captures the spatial configuration of the scene.

SfM points are particularly beneficial for Audio-Visual (AV) reconstruction due to two key factors: 1) They provide detailed 3D scene geometry, which is essential for spatial information in both visual and audio rendering. The points capture physical boundaries and surfaces in the scene from which sound waves can reflect, diffract, or be absorbed. This geometric data is critical for creating realistic audio-visual scenes and can be reused by any emitter or listener within the environment. 2) SfM points can also be used as starting points for visual renderer such as 3D-GS [3], facilitating the parallel synchronization of audio and visual rendering. In our work, given training videos, we employ COLMAP [1, 2] to estimate the SfM points together with the camera calibration.

## 3.2 Audio-Visual Anchors

We use Audio-Visual Anchors as basic units for construction of point-based audio-visual scene representation. The anchors are clustered from SfM points described in Section 3.1. However, there are often multiple (tens of thousands) of raw SfM points with less than 0.5m of each other. This density is too high given that grid resolutions for room acoustic tasks are usually larger than 0.5m [32, 45, 42]. For efficiency, we first merge points within every 0.25m, averaging their RGB values to form a single grid. We then select representative $N$ anchors from the initial SfM points using K-Means based on their location distribution. After clustering, we initialize the anchor features as follows: 1) **3D coordinates**: Each cluster center $(x_i, y_i, z_i)$ is taken as the 3D coordinate of the corresponding anchor; 2) **RGB feature**: The RGB values of the $K$ nearest points are concatenated to form the initial RGB feature, with a dimension of $\mathbb{R}^{N \times K \times 3}$; and 3) **Latent Audio Embedding**: A latent audio embedding $\mathbf{e}_i \in \mathbb{R}^C$ is used to represent audio effects. This embedding captures how the anchor region contributes to sound propagation from different listener viewpoints.

## 3.3 Audio-Visual Cloud Splatting (AVCS)

We introduce the Audio-Visual Cloud Splatting (AVCS) module for decoding audio spatial effect transfer function from a point-based scene representation given any listener viewpoint. As illustrated in Figure 2, AVCS involves two primary components: 1) Anchor Projection and 2) Visual-to-Audio Splatting Transformer. At the onset, each Audio-Visual Anchor is projected to the head coordinate system of the target listener, followed by the integration of anchor features for each audio frequency band using a Visual-to-Audio Splatting Transformer. The transformer outputs two acoustic masks: a mixture mask $\mathbf{m}_m$ and a difference mask $\mathbf{m}_d$. The masks act as transfer functions to convert the monaural reference sound into stereo audio at the viewpoint of the listener.

**Anchor Projection**

We define the viewpoint pose as $\{\mathbf{R} = (R_x, R_y, R_z), \mathbf{T} = (T_x, T_y, T_z)\}$, where $\mathbf{R}$ and $\mathbf{T}$ represent the rotation matrix and translation vector in the 3D world coordinate system, respectively. Given Anchor $i$ located at $\mathbf{p}_i = (x_i, y_i, z_i)$ in the world coordinate system, we calculate the projected Relative Vector $\mathbf{r}_i \in \mathbb{R}^C$ in the listener head coordinate system:

$$r_i = \Phi(\mathbf{R} \cdot (\mathbf{p}_i - \mathbf{T})), \tag{1}$$

where $\Phi$ represents positional encoding applied to obtain the higher-dimensional embedding of $\mathbf{r}_i$.

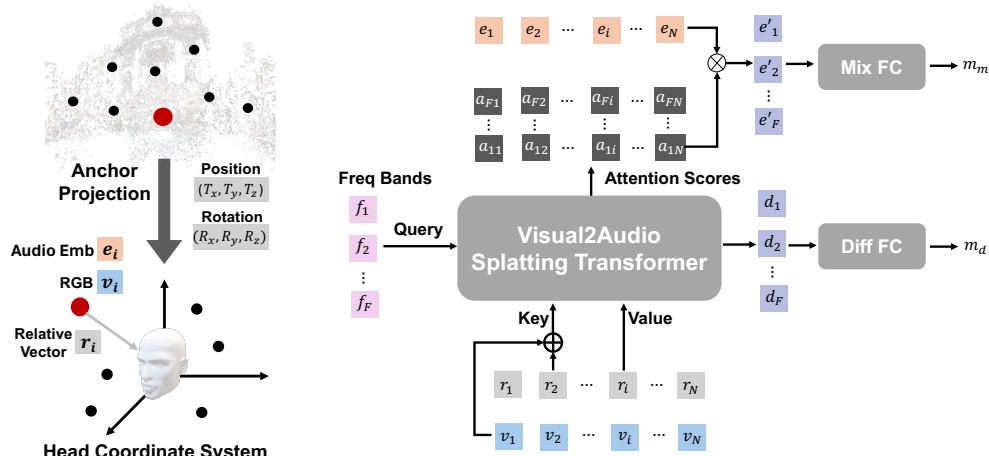

Figure 2: AVCS consists of two components: Anchor Projection (left) and Visual-to-Audio Splatting Transformer (right). Audio-Visual Anchors are projected into the coordinate system of the listener head, and the transformer decodes features for each audio frequency band, outputting two acoustic masks to convert the monaural reference sound into stereo audio at the target viewpoint.

**Visual-to-Audio Splatting Transformer**

Given the emitter, sound effects can vary with viewpoint poses and sound frequencies due to differences in sound propagation process. Considering this, we propose the Visual-to-Audio Splatting Transformer to dynamically weigh the contribution of each anchor by frequencies and view poses.

The input to the transformer consists of **key**, **query**, and **value** sequences. Specifically, the **queries** are embedded indices $\mathbf{f}_k \in \mathbb{R}^{F \times C} | k = 1, 2, ..., F$ corresponding to $F$ frequency bands. The **keys** consist of a combination of the RGB visual features of each anchor and its relative vector $\mathbf{r}_i$ in the listener's head coordinate system, shaped as $\mathbb{R}^{N \times C}$. These visual features are to enhance scene understanding and distinguish between anchors at different locations. A two-layer multi-layer perceptron (MLP) is used to reduce the dimensionality of initial RGB features from $\mathbb{R}^{K \times 3}$ to $\mathbf{v}_i \in \mathbb{R}^C$ of each anchor. The **values** are also composed of the relative vector $\mathbf{r}_i$, similarly shaped as $\mathbb{R}^{N \times C}$.

As depicted in Figure 2 (right), for frequency band $k = \{1, 2, ..., F\}$, the decoding process of the Visual-to-Audio Splatting Transformer layer is formulated as

$$a_{ki} = softmax(\frac{\mathbf{f}_k \cdot (\mathbf{r}_i + \mathbf{v}_i)^T}{\sqrt{C}})(i = 1, 2, ..., N), \quad \mathbf{d}_k = \sum_{i=1}^{N} a_{ki}\mathbf{r}_i, \quad \mathbf{e'}_k = \sum_{i=1}^{N} a_{ki}\mathbf{e}_i, \quad (2)$$

where the *softmax* function normalizes the contribution of each anchor.

The output of the transformer consists of: 1) An **Attention Mask** $a_{ki}, \in \mathbb{R}^{F \times N}$, which indicates the contribution weight of each anchor and represents the influence of each anchor on the spatial audio effect across frequency bands (higher weights indicate greater influence, examples shown in Figure 6); and 2) **integrated Relative Vector embedding** of the anchors for each frequency band w.r.t the target viewpoint pose, shaped as $\mathbb{R}^{F \times C}$, which is closely related to the target head orientation.

The decoding process dynamically "enhances" anchors into the frequency domain for varying viewpoints by applying the attention score $a_{ki}$ as weights to the audio embedding $\mathbf{e}_i$ (obtaining the mixture Audio Embedding $\mathbf{e'}$) and the Relative Vector embedding $\mathbf{r}_i$ (obtaining integrated Relative Vector embedding $\mathbf{d}$), as depicted in Equation 2. This results in a transfer function to manipulate spatial sound effects through two acoustic masks. The first mask, the mixture mask $\mathbf{m}_m \in \mathbb{R}^F$, is derived from the mixture Audio Embedding $\mathbf{e'} \in \mathbb{R}^{F \times C}$ and reflects changes in the audio magnitude of the output mixture sound. The second mask, the difference mask $\mathbf{m}_d \in \mathbb{R}^F$, is computed from the integrated Relative Vector embedding $\mathbf{d} \in \mathbb{R}^{F \times C}$. The difference mask is sensitive to the head orientation of the listener and affects the energy distribution difference of stereo channels.

$$\mathbf{m}_m = FC_m(\mathbf{e'}), \quad \mathbf{m}_d = 2 * \sigma(FC_d(\mathbf{d})) - 1, \quad (3)$$

where $FC_m$ and $FC_d$ indicate the Fully Connected Layers, $\sigma$ represents the sigmoid function used to constrain the range of the left-right spectrogram magnitude difference.

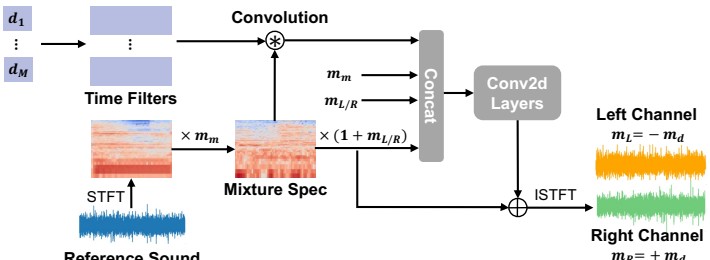

Figure 3: SARH implements a single-layer residual structure to transform monaural reference sound into stereo audio. It contains two convolution modules: Time Filters and Conv2D layers to adjust energy distribution in both the time and frequency domain and enhance the stereo output.

## 3.4 Spatial Audio Render Head (SARH)

After obtaining the acoustic masks from the Visual-to-Audio Splatting Transformer, we design the Spatial Audio Render Head (SARH) as illustrated in Figure 3. SARH transfers the input monaural reference sound into stereo audio. We first apply the Short-Time Fourier Transform (STFT) to the reference sound, extracting its spectrogram $\mathbf{S} \in \mathbb{R}^{F \times T}$. To generate stereo audio, we calculate the left $\mathbf{S}_L \in \mathbb{R}^{F \times T}$ and right channel spectrogram outputs $\mathbf{S}_R \in \mathbb{R}^{F \times T}$ as follows:

$$\mathbf{S}_m = \mathbf{m}_m\mathbf{S}, \quad \mathbf{S}_L = (1 - \mathbf{m}_d)\mathbf{S}_m, \quad \mathbf{S}_R = (1 + \mathbf{m}_d)\mathbf{S}_m \tag{4}$$

SARH utilizes a single-layer residual structure, which contains two convolution modules in the residual unit: **Time Filters** and **Conv2D Layers**. These modules adjust the energy distribution across both the time and frequency domains to enhance the stereo output.

**Time Filters.** Given that the reference sound may have distinct acoustic properties, such as reverberation time, the first convolution module, Time Filters, adjusts the energy distribution of the mixture output spectrogram $\mathbf{S}_m$ across the time domain, for more accurate acoustic effects and higher sound quality. We condition on the integrated Relative Vector embedding $\mathbf{d}$ to generate convolution kernels and biases of the filters, where the input and output channel dimension of filters are both the time window number $T$ of the spectrogram. An alternative approach utilizes an attention decoder layer, similar to that described in Section 3.3. Here, the query consists of embedded indices $\{\mathbf{t}_k \in \mathbb{R}^{T \times C} | k = 1, 2, ..., T\}$ of $T$ time windows. The key is determined by Spherical Harmonics parameters, which are based on the vector from each anchor to the viewpoint in the world coordinate system (head orientation is not considered for these time domain adjustments on the mixture spectrogram). The value is latent Time Filter Embedding of each anchor which decodes into a convolution filter kernel with the shape of $\mathbb{R}^{T \times T}$ and bias $\mathbb{R}^{F \times T}$. This setting reduces parameters while still keeps competitive accuracy, as detailed in the experiments in Section 4.2.

**Conv2D Layers.** After processing through the Time Filters, Conv2D Layers smooth and enhance the time-frequency domain energy distribution. To obtain the inputs, the new mixture output spectrogram $\mathbf{S}'_m$ is concatenated with $\{\mathbf{m}_m, -\mathbf{m}_d$ and $\mathbf{S}_L\}$ for the left channel, $\{\mathbf{m}_m, \mathbf{m}_d$ and $\mathbf{S}_R\}$ for the right channel, respectively. A lightweight stacked Conv2D layer network then processes these inputs to refine the time-frequency energy distribution and generates $\mathbf{S}'_L$ and $\mathbf{S}'_R$. At the final stage, the original stereo output spectrograms $\mathbf{S}_L$ and $\mathbf{S}_R$ are added to $\mathbf{S}'_L$ and $\mathbf{S}'_R$ via skip connections respectively to obtain final output $\tilde{\mathbf{S}}_L$ and $\tilde{\mathbf{S}}_R$, as shown in Equation 4.

## 3.5 Training

We implement an end-to-end training strategy to minimize the discrepancy between the rendered and target stereo spectrograms. To calculate the loss, we first encode the ground truth stereo audio using the Short-Time Fourier Transform (STFT) to spectrograms and then apply the Mean Squared Error (MSE) loss between the logarithm of the predicted spectrogram and the ground-truth (GT) spectrogram magnitudes on both the left and right channels. The loss function is of the form

$$\mathcal{L} = \frac{1}{2}\left(\text{MSE}(\log|\tilde{\mathbf{S}}_L|, \log|\hat{\mathbf{S}}_L|) + \text{MSE}(\log|\tilde{\mathbf{S}}_R|, \log|\hat{\mathbf{S}}_R|)\right), \tag{5}$$

where $\tilde{\mathbf{S}}_L$ and $\tilde{\mathbf{S}}_R$ are predicted stereo spectrograms, $\hat{\mathbf{S}}_L$ and $\hat{\mathbf{S}}_R$ are GT stereo spectrograms. As in [46, 29], we exclusively focus on the distance of the magnitude values and do not supervise exact phase values in order to preserve higher audio quality of the output.

# 4 Experiments

## 4.1 Datasets and Metrics

**Datasets** This work focuses on audio-visual synthesis for real-world scenes. Real-world data present challenges such as background noise and discrepancies in sound propagation compared to simulated environments. We have conducted experiments on the following two real-world datasets.

1) **RWAVS Dataset** [29]: RWAVS includes audio-visual data from 13 everyday indoor and outdoor settings featuring multiple viewpoints across diverse environments. Each recording captures both visual and auditory data with key frames captured at 1 frame per second (FPS). Camera poses are estimated using COLMAP [47], and each key frame is paired with 1 sec of binaural audio and source audio. The dataset is considered particularly challenging due to its diverse environments and varying camera poses. As in [29], we split 80% data as training samples and the rest for validation, with all audio resampled to a frequency of 22050 Hz. We optimize a single model per scene for all methods and report the average results across all scenes in Table 1.

2) **Replay-NVAS Dataset** [46]: Replay-NVAS consists of multi-view recordings of scenes within a single apartment, capturing 46 different daily scenarios such as conversations, dinners and yoga sessions from 8 viewpoints with captured stereo sounds. Each human participant wears a near-range microphone to record their clean speech audio which serves as reference sound in our experiments. This dataset is challenging due to its wide range of social activities, ambient noise, room reverberations, overlapping speech and dynamic human performers. As in [46], we apply bandpass filtering to cut frequencies below 150 Hz to reduce acoustic variations among viewpoints and resample the audio samples to a frequency of 16000 Hz. 28/6/7 multi-view videos are used for training, validation and testing, respectively. We use COLMAP to estimate camera poses together for all videos.

**Metrics** We evaluate audio reconstruction performance using five key metrics **(the lower the better)**. 1) Magnitude Spectrogram Distance (**MAG**) [46] assesses the closeness of the reconstructed audio to the ground truth by measuring the distance in the magnitude spectrogram. 2) Left-Right Energy Ratio Error (**LRE**) [46] evaluates the accuracy of spatial sound by calculating the difference of ratio of the energy between left and right channels. 3) Energy Envelope Error (**ENV**) [24] assesses the Euclidean distance between the energy envelopes of the groundtruth and predicted left and right audio waveform channels. 4) RT60 Error (**RTE**) [22, 23] quantifies inaccuracies in the predicted reverberation time as it decays by 60dB using a pretrained model for estimation. 5) Deep Perceptual Audio Metric (**DPAM**) [48], a deep learning-based perceptual quality metric aligned with human judgments. These metrics span various aspects, from spectrogram accuracy to spatial and acoustic properties, integrating both technical and perceptual elements of sound quality. Furthermore, we report the speed results in FPS in Tables 1. Speed tests are conducted on a GeForce RTX 2080 Ti, with results averaged over 1000 samples.

## 4.2 Comparison with Baselines

We compare our method with the following baseline approaches: 1) **MonoMono** [29]: Duplicates the source audio to synthesize binaural audio. 2) **Mono-Energy** [29]: Scales input audio to match the known average energy of the target audio and duplicates it to stereo channels. 3) **Stereo-Energy** [29]: Scales the energy of the input audio for each channel based on the known energy levels of the target audio. 4) Digital Signal Processing (**DSP**) [49, 46]: Utilizes digital signal processing to adjust audio based on the distance of the source, azimuth, and elevation, applying head-related transfer functions (HRTFs) to estimate the spatial audio at the target microphone location. 5) **ViGAS** [46]: Transforms the sound to the target viewpoint by reasoning about the observed audio and visual stream. 6) **VAM** [22]: Matches the acoustics of input audio with a target image that is adapted by incorporating the image from the source viewpoint with the target viewpoint pose. 7) **AV-NeRF** [29]: A NeRF-based system that synthesizes binaural audio for a given camera pose by first rendering a pair of RGB and depth images from the same camera position. 8) **NACF** [43]: Neural Acoustic Context Field that parameterizes an audio scene by incorporating multiple acoustic contexts such as geometry, material properties, and spatial information, adapted here to predict waveform masks to render target audio. 9) **INRAS** [32]: Uses implicit neural fields to disentangle and represent audio scenes for waveform masks prediction. 10) **NAF** [42]: Employs local feature grids and an implicit decoder to model sound propagation in physical scenes, modified to predict magnitude masks on time-frequency domain. For *NACF*, *INRAS* and *NAF* we employ the same set of Audio-Visual Anchors in place of the grids that has been traditionally used in these methods for fair comparison. We report main

| Dataset | Methods | # Params | FPS | Image | MAG | LRE | ENV | RTE | DPAM |
|---|---|---|---|---|---|---|---|---|---|
| RWAVS [29] | Mono-Mono | - | - | ✗ | 1.460 | 1.328 | 0.445 | 0.132 | 0.756 |
| | Mono-Energy | - | - | ✗ | 0.532 | 1.328 | 0.156 | 0.145 | 0.510 |
| | Stereo-Energy | - | - | ✗ | 0.560 | - | 0.160 | 0.143 | 0.535 |
| | DSP [50] | 163M | - | ✗ | 1.016 | 3.468 | 0.274 | 0.119 | 0.588 |
| | VAM [51] | 46.7M | 66 | ✓ | 0.390 | 0.996 | 0.156 | 0.079 | 0.459 |
| | ViGAS [28] | 13.1M | 34 | ✓ | 0.370 | 1.089 | _0.147_ | 0.094 | 0.357 |
| | AVNeRF [29] | 12.0M | 119 | ✓ | 0.370 | 1.013 | **0.145** | 0.098 | 0.381 |
| | NACF [43] | 0.44M | 41 | ✗ | 0.459 | 1.364 | 0.176 | 0.138 | 0.506 |
| | INRAS [32] | 0.31M | 180 | ✗ | 0.455 | 1.503 | 0.179 | 0.148 | 0.485 |
| | NAF [42] | 0.22M | 99 | ✗ | 0.448 | 1.204 | 0.522 | 0.138 | 0.353 |
| | AV-Cloud (Ours) | 3.91M | 83 | ✗ | **0.351** | **0.936** | 0.145 | _0.074_ | **0.276** |
| | AV-Cloud-SH | 1.29M | 171 | ✗ | _0.355_ | _0.983_ | _0.147_ | **0.073** | **0.276** |
| | AV-Cloud-sim-SH | 0.51M | 301 | ✗ | 0.359 | 0.996 | _0.147_ | 0.076 | _0.291_ |
| Replay-NVAS [28] | Mono-Mono | - | - | ✗ | 0.313 | 0.934 | 0.127 | 0.366 | 0.521 |
| | Mono-Energy | - | - | ✗ | 0.191 | 0.934 | **0.050** | 0.356 | 0.496 |
| | Stereo-Energy | - | - | ✗ | 0.196 | - | 0.054 | 0.357 | 0.473 |
| | DSP [50] | 163M | - | ✗ | 0.228 | 6.186 | 0.066 | 0.338 | 0.482 |
| | VAM [51] | 46.5M | 66 | ✓ | 0.239 | 0.824 | 0.062 | 0.147 | 0.458 |
| | ViGAS[28] | 12.7M | 34 | ✓ | 0.193 | 0.698 | 0.054 | 0.137 | 1.177 |
| | AVNeRF [29] | 11.8M | 119 | ✓ | 0.214 | 0.773 | 0.055 | 0.159 | 0.290 |
| | NACF [43] | 0.54M | 45 | ✗ | 0.298 | 0.722 | 0.079 | 0.332 | 0.544 |
| | INRAS [32] | 0.32M | 162 | ✗ | 0.211 | 0.928 | 0.058 | 0.340 | 0.807 |
| | NAF [42] | 0.23M | 100 | ✗ | 0.208 | 0.820 | 0.059 | 0.388 | 0.565 |
| | AV-Cloud (Ours) | 2.47M | 103 | ✗ | **0.180** | **0.600** | _0.052_ | _0.065_ | _0.234_ |
| | AV-Cloud-SH | 1.24M | 133 | ✗ | _0.181_ | _0.673_ | 0.053 | _0.065_ | 0.244 |
| | AV-Cloud-sim-SH | 0.48M | 207 | ✗ | **0.180** | 0.689 | _0.052_ | **0.062** | **0.231** |

Table 1: Comparison of AV-Cloud with state-of-the-art methods. For each metric, the top1 value is highlighted in bold, the second best is underlined, lower is better.

comparison results on the RWAVS and Replay-NVAS datasets in Table 1. The implementation details of AV-Cloud are introduced in Appendix A.1.

Our experiments results summarized in Table 1 show that AV-Cloud significantly surpasses baselines across the acoustic metrics, even with relatively fewer model parameters and higher inference speed. Non-learning methods such as *Mono-Energy* and *Stereo-Energy*, that assume prior knowledge of the target sound energy distribution, perform well on metrics such as ENV and LRE, but are impractical for real-world applications due to their reliance on unavailable data. The *DSP* struggles due to difficulties in estimating precise distances and scene-specific variations in HRTF functions. Image-based methods such as *VAM* [22], *ViGAS* [46] and *AVNeRF* [29] necessitate extensive model parameters and depend on the visual rendering results. Point-based methods such as *NACF* [43], *INRAS* [32] and *NAF* [42] construct implicit neural representations of scenes but do not account for how points contribute across viewpoints. This results in less effective representation and generalization capabilities. In contrast, *AV-Cloud* leverages Audio-Visual Anchors clustered from sparse SfM points and dynamically reweighs them to enhance spatial audio accuracy. *AV-Cloud* enhances the audio rendering metrics such as MAG (>5%), LRE (>6%), and perceptual quality DPAM (>19%), outperforming even image-based methods. Additionally, our Spatial Render Head adjusts reverberation effects through Time Filters and achieves more accurate outcomes on the acoustics represented by the metric RTE. Specifically for the challenging indoor scenario Replay-NVAS set, *AV-Cloud* reduces RTE by 53% relative to the second best method *ViGAS*.

As outlined in Section 3.4, we also propose an alternative approach (*AV-Cloud-SH*) with an attention decoder layer to derive Time Filters, with keys based on Spherical Harmonics (SH) parameters. This strategy cuts parameter count by >50% and at the same time it maintains competitive accuracy. To reduce the complexity of the model, we developed the Visual-to-Audio Splatting Transformer (Section 3.3) using a single attention decoder layer (*AV-Cloud-sim-SH*), resulting in 0.51M parameters. This is comparable to other point-based methods such as *NACF*, *INRAS* and *NAF* but achieves inference speeds approximately 30% faster. When *AV-Cloud-sim-SH* is compared to these three methods on RWAVS validation set, it enhances the accuracy of MAG and LRE by 20% and 17% respectively, and significantly reduces the acoustic reverberation error RTE by 45%, and improves perceptual quality, DPAM, by 18%.

| Methods | MAG | LRE | ENV | RTE | DPAM |
|---------|-----|-----|-----|-----|------|
| AV-Cloud | **0.351** | **0.936** | **0.145** | 0.074 | **0.276** |
| baseline | 0.488 | 1.329 | 0.240 | 0.130 | 0.357 |
| w/o AVCS | 0.368 | 1.124 | 0.150 | 0.080 | 0.318 |
| w/o time | 0.356 | 0.984 | 0.146 | 0.084 | 0.281 |
| w/o a_emb | 0.354 | 0.975 | 0.146 | 0.072 | 0.282 |
| w/o rgb | 0.361 | 1.014 | 0.148 | 0.077 | 0.291 |
| w/o 2 masks | 0.357 | 0.983 | 0.147 | **0.070** | 0.280 |
| w/o conv2d | 0.359 | 1.019 | 0.148 | 0.074 | 0.280 |

Table 2: **Ablations on RWAVS validation set.**

| Methods | NAF | AVNeRF | AV-Cloud |
|---------|-----|--------|----------|
| Votes | 15% | 37% | **48%** |

Table 3: **Human Study for In-the-Wild Experiments.** AV-Cloud is preferred over the other two methods by a large margin.

## 4.3 Ablation Studies

We conducted ablation studies on RWAVS validation set to verify the contribution of key components.

**Baseline.** We use an MLP to predict mixture and difference masks across all frequency bands, based on the viewpoint pose, and render the target spatial sound using Equation 4 as our *baseline*. Compared to the full *AV-Cloud* and other variants in Table 2, the *baseline* does not perform well across all metrics. This indicates limited generalization to novel views when solely considering viewpoint poses.

**Audio-Visual Cloud Splatting (AVCS).** To verify the contribution of AVCS module in Section 3.3, we implement a variant *w/o AVCS* that replaces the AVCS module with an MLP as *baseline* to generate the mixture and difference masks directly from viewpoint poses. It appears that *w/o AVCS* struggles to capture the relationship between left-right channel energy and viewpoint poses, reducing the LRE accuracy by 20% comparing to full *AV-Cloud*. The AVCS module dynamically adjusts anchor contributions for audio rendering based on the viewpoint of the listener which results in more accurate spatial audio effects. Example attention weights of AV anchors are visualized in Figure 6.

**Visual Feature Contribution.** To evaluate the impact of visual features, we remove the RGB features from the Visual-to-Audio Splatting Transformer, as reported in Section 3.3 as *w/o rgb*. This change decreases perceptual quality by 5%, spatial effect accuracy LRE by 8%, and increases the MAG error from 0.351 to 0.361. It demonstrates that RGB features can enhance anchor distinction and improve the ability of the attention mechanism to weigh contributions based on scene understanding.

**Audio Embedding.** The latent Audio Embedding $e_i$ for each Audio-Visual Anchor adjusts the audio mix based on distance and acoustic effects between the emitters and the listener. Removing $e_i$ (*w/o a_emb* in Table 2) and using only the Relative Vector embedding $d$ to predict the mixture mask $m_m$, similarly to the difference mask $m_d$, decreases audio rendering accuracy. While this variant slightly improves the RTE accuracy (error -0.002) it reduces the accuracy of the spatial effect metric LRE by 4%. The difference mask and mixture mask target distinct audio factors, with the mixture mask emphasizing scene properties and listener-emitter locations. The difference mask focuses on the head orientation. Thus decoupling the prediction with two masks and Audio Embedding is more effective.

**Spatial Audio Render Head (SARH).** Our results of comparing the *baseline* and *w/o AVCS* demonstrate that the use of the residual convolution module of SARH significantly improves rendering accuracy: MAG by 25%, ENV by 38%, RTE by 38%, and DPAM by 11% (Table 2). In the *w/o time* variant, where only the 2D convolutional layer is retained, reverberation time error increases from 0.074 to 0.084, and LRE error increases by 5%, demonstrating the key role of Time Filters in adjusting the time domain energy distribution, which in turn impacts reverberation time and overall acoustic quality. The *w/o 2 masks* variant, where a direct prediction of the two-channel spatial effect transfer mask is utilized instead of predicting the difference and mixture masks separately, results in an increase in error metrics, particularly notable for LRE. Similarly, for the *w/o conv2d* variant, where the Conv2D layers are removed, relying solely on Time Filters, the error metrics also show an increase. These ablations highlight the significant contribution of the SARH module, emphasizing the significant roles of its key components, Time Filters and Conv2D layers. SARH along with the residual module improve rendering accuracy and overall audio quality.

## 4.4 Qualitative Results Comparison

In Figure 4, we illustrate the rendered spectrogram (for mixture sound) and stereo waveforms compared across the methods along with input and GT. The region circled in blue on the spectrogram highlights the location of the reverberation effect in the spectrogram. For indoor scenes reverberation

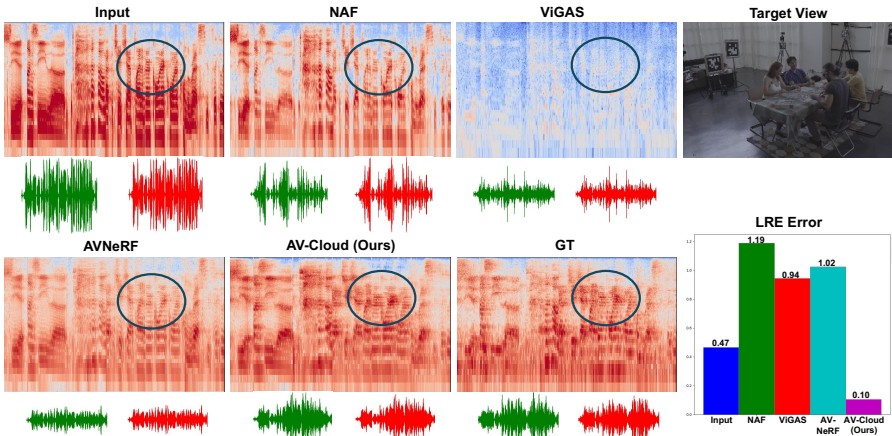

Figure 4: **Qualitative Results Comparisons.** Left: output spectrogram and stereo waveforms; Bottom Right: LRE error bar chart. The blue-circled region on the spectrogram highlights the reverberation effect, demonstrating ability of AV-Cloud to capture prolonged energy decay.

indeed has a longer energy decay due to wall reflections. As can be observed from comparing the spectrogram and GT, methods significantly vary in their ability to reproduce the reverberation. Specifically, *NAF* does not convey reverberation from input to target sound, *AVNeRF* shows better rendering but is not accurate enough in reverberation and spectrogram magnitude. Both methods produce waveforms that differ significantly from GT. *ViGAS* renders a more accurate waveform but lacks precision in reverberation and perceptual quality. AV-Cloud turns out to reproduce the effect closer to GT. This can be also seen from the barchart depicting the LRE error (bottom right) which indicates such spatial effect as reverberation. LRE value for AV-Cloud is significantly lower in comparison to other methods.

### 4.5 In-the-Wild Experiments

We tested our method in a real-world virtual tourism scenario in a scene which includes a fountain. We generated audio samples for five different routes and conducted a human study with 76 participants, who watched navigation videos featuring changing viewpoints and spatial audio rendered by *NAF*, *AVNeRF*, and our *AV-Cloud*, and then chose the video which sound best matched the visual perspectives. Participants were instructed to select the video where spatial audio matches the visual content, focusing on left-right ear effects and overall synchronization. As shown in Table 3, *AV-Cloud* received a majority vote of 48%, outperforming *AVNeRF* and *NAF*. Furthermore, we developed a webGL-based platform that enables real-time audio-visual rendering based on view perspectives with an Apple M2 chip. More details can be found in the Appendix A.2.

## 5 Conclusion

In this work, we introduced a novel point-based audio-visual rendering framework, AV-Cloud, which allows for high-quality audio rendering from any viewpoint without the need for pre-rendered images and is synchronized with visual rendering. AV-Cloud is based on Audio-Visual Anchors that enhance scene representation. The Audio-Visual Cloud Splatting module dynamically adjusts anchor contributions conditioned on the target viewpoint. Experiments show that AV-Cloud achieves higher accuracy and reliability in spatial audio rendering and outperforms existing methods in real-world experiments. We also discuss the limitations and broader impacts of our work in Section A.9 and A.10 respectively.

## Acknowledgments and Disclosure of Funding

We acknowledge the support of HDR Institute: Accelerated AI Algorithms for Data-Driven Discovery (A3D3) National Science Foundation grant PHY-2117997 and the departments of Applied Mathematics and Electrical and Computer Engineering at the University of Washington.

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

# A Appendix

## A.1 Implementation Details

For all experiments, we utilize COLMAP [47] to derive initial SfM points and camera pose estimation from the provided videos for each scene. We set the K-Means cluster number $N$ to 256, and initialize each of these anchors with RGB values of the nearest $K = 50$ points in Section 3.2. We utilize the Short-Time Fourier Transform (STFT) to convert waveform audio into the time-frequency domain, setting the FFT size, window length, and hop length to 512, 512, and 128 respectively, and applying a Hanning window with it. For the Visual-to-Audio Splatting Transformer (Section 3.3), we deploy a 3-layer attention module, set the frequency band number $F$ and embedding dimension $C$ to 257 and 128, respectively. Time Filters (Section 3.4) are developed using distinct 2-layer 1D convolution modules that generate filter kernels and biases, conditioned on the integrated Relative Vector. The convolution for time distribution adjustments has a kernel size of 3 on the frequency domain. For implementation of Spherical Harmonics (SH) parameters, we use 1-degree SH to create Time Filters with a convolution kernel size of 1 in the frequency domain. The Conv2d Layers in the render head's residual unit (Section 3.4) which comprise three stacked Conv2d modules with kernel sizes of 7, 3, and 3, and a hidden channel size of 16. We employ the Adam optimizer for optimization, using an exponentially decaying rate starting from 0.01 and spanning over 100 epochs, with a batch size of 6.

For the RWAVS dataset, *AVNeRF* [29] utilizes explicit emitter locations for coordinate transformations [29], while emitter locations are not given as input for the remaining methods. For Replay-NVAS dataset, for image-based methods, bounding boxes are used to identify active speakers when multiple sources are present [22, 28, 29]. For methods that do not utilize image data such as [32, 42] and ours, the intrinsic matrix is used to derive view vectors in the world coordinate system. These vectors provide clues about which participant is currently speaking, and are used in our method to enhance the Relative Vector for each anchor.

## A.2 Details of In-the-Wild Experiments

We conducted in-the-wild virtual tourism scenario experiments using videos collected across four different routes around a fountain, captured with an iPhone 14 Pro camera. Our model was trained on the combined 1-minute-long video collection, using a 44,100 Hz audio sampling rate and 0.5-second audio-visual clips per training sample with 5% overlap. Following the same settings as in Section A.1, the model was trained for 100 epochs. Testing was conducted on five unseen routes with continuously changing viewpoints. Generic overlap-add with linear fade-in/fade-out between adjacent test samples was used to smooth the output spatial audio during viewpoint transitions. Please refer to our attached videos to see samples of the testing results.

To facilitate navigation of rendered 3D audio-visual scenes, we developed a WebGL-based audio-visual navigation platform using using JavaScript and HTML based on [52]. [52] is a WebGL implementation of a real-time renderer for 3D Gaussian Splatting [3] for Real-Time Radiance Field Rendering. We utilize [52] to render the 3D visual scene, while audio was generated directly from the Audio-Visual Anchors (defined in Section 3.2) in parallel with visual rendering, enabling real-time rendering of audio-visual scenes according to viewpoint and perspective. To our knowledge, this is the first deep learning-based 3D audio-visual navigation platform that can run real-time with WebGL. Deployed in our navigation platform, AV-Cloud can achieve over 25 FPS on audio rendering (each sample with 257 spectrogram frequencies and 182 time frames i.e. 0.5s of audio sampled in 44,100 Hz for our experiments) on an Apple M2 chip.

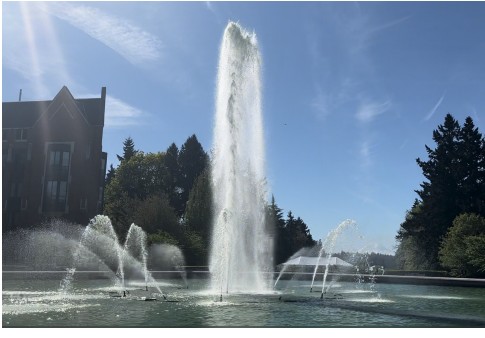
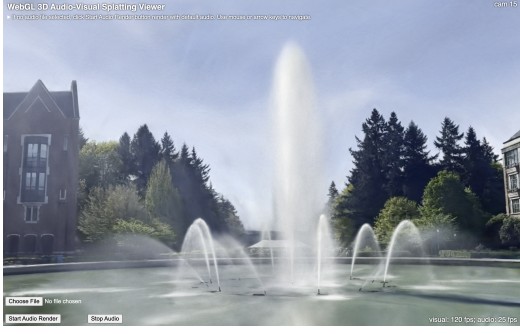

| (a) Training Sample for Fountain Scenario. | (b) Figure of our Audio-Visual Navigation Platform. |

Figure 5: Example of training sample for the fountain scenario (a) and a screenshot of our WebGL-based Audio-Visual Navigation Platform (b). The platform enables navigating the scene while the spatial audio adapts to viewpoint perspective changes.

## A.3 Generated Samples

Please refer to the attached videos in the supplementary **demo_videos** folder for samples of video clips generated by our model. Please ensure that your audio is turned on and use headphones or speakers with **stereo audio capability** for the best experience. For the RWAVS samples, we show the visualization of navigation route and Audio-Visual Anchors (anchor radius expands linearly with the attention weights) in the right. Please see more details about visualization and interpretation in Section A.7.

## A.4 Detailed Performance Results on RWAVS Scenes

| Methods | Office | | | | | House | | | | |
| --- | MAG | LRE | ENV | RTE | DPAM | MAG | LRE | ENV | RTE | DPAM |
| --- | --- | --- | --- | --- | --- | --- | --- | --- | --- | --- |
| Mono-Mono | 1.400 | 1.901 | 0.415 | 0.126 | 0.750 | 1.430 | 0.886 | 0.422 | 0.144 | 0.855 |
| Mono-Energy | 0.465 | 1.901 | 0.140 | 0.129 | 0.534 | 0.599 | 0.886 | 0.174 | 0.148 | 0.629 |
| Stereo-Energy | 0.479 | - | 0.142 | 0.128 | 0.558 | 0.632 | - | 0.179 | 0.147 | 0.656 |
| DSP [50] | 0.670 | 4.379 | 0.196 | 0.110 | 0.531 | 0.959 | 3.913 | 0.262 | 0.117 | 0.703 |
| VAM [51] | 0.364 | 1.477 | 0.142 | **0.087** | 0.461 | 0.392 | 0.822 | 0.161 | 0.063 | 0.511 |
| INRAS [32] | 0.369 | 1.916 | 0.156 | 0.146 | 0.385 | 0.498 | 1.356 | 0.194 | 0.167 | 0.601 |
| NAF [42] | 0.331 | 1.669 | 0.178 | 0.120 | **0.271** | 0.520 | 0.941 | 0.475 | 0.160 | 0.426 |
| ViGAS [28] | 0.308 | 1.548 | **0.128** | 0.109 | 0.344 | 0.394 | 0.957 | 0.160 | 0.071 | 0.461 |
| AVNeRF [29] | 0.305 | 1.430 | 0.130 | 0.125 | 0.293 | 0.401 | 0.889 | **0.154** | 0.071 | 0.463 |
| AV-Cloud (Ours) | **0.300** | **1.388** | 0.131 | 0.093 | 0.281 | **0.382** | **0.797** | 0.156 | **0.059** | **0.289** |
| AV-Cloud-SH | 0.302 | 1.506 | 0.131 | 0.095 | 0.275 | 0.386 | 0.830 | 0.160 | 0.061 | 0.302 |
| AV-Cloud-sim-SH | 0.303 | 1.471 | 0.130 | 0.092 | 0.277 | 0.388 | 0.818 | 0.159 | 0.062 | 0.325 |

| Methods | Apartment | | | | | Outdoor | | | | |
| --- | MAG | LRE | ENV | RTE | DPAM | MAG | LRE | ENV | RTE | DPAM |
| --- | --- | --- | --- | --- | --- | --- | --- | --- | --- | --- |
| Mono-Mono | 1.476 | 0.863 | 0.472 | 0.142 | 0.595 | 1.535 | 1.663 | 0.470 | 0.115 | 0.824 |
| Mono-Energy | 0.657 | 0.863 | 0.188 | 0.181 | 0.331 | 0.409 | 1.663 | 0.121 | 0.121 | 0.545 |
| Stereo-Energy | 0.690 | - | 0.194 | 0.177 | 0.352 | 0.438 | - | 0.124 | 0.121 | 0.573 |
| DSP [50] | 0.917 | 3.344 | 0.253 | 0.128 | 0.400 | 1.519 | 2.235 | 0.386 | 0.120 | 0.718 |
| VAM [51] | 0.504 | 0.768 | 0.198 | 0.076 | 0.352 | 0.298 | 0.919 | 0.124 | 0.090 | 0.512 |
| INRAS [32] | 0.614 | 0.830 | 0.220 | 0.154 | 0.419 | 0.338 | 1.911 | 0.147 | 0.124 | 0.534 |
| NAF [42] | 0.645 | 1.003 | 1.058 | 0.171 | 0.313 | 0.297 | 1.201 | 0.377 | 0.100 | 0.403 |
| ViGAS [28] | 0.524 | 0.942 | 0.190 | 0.109 | 0.257 | 0.254 | 0.908 | 0.111 | 0.089 | 0.368 |
| AVNeRF [29] | 0.514 | 0.832 | 0.188 | 0.110 | 0.377 | 0.258 | 0.900 | 0.108 | 0.084 | 0.391 |
| AV-Cloud (Ours) | **0.483** | **0.751** | **0.185** | **0.065** | **0.228** | **0.238** | 0.809 | **0.107** | 0.079 | 0.305 |
| AV-Cloud-SH | 0.489 | 0.804 | 0.187 | **0.065** | **0.228** | 0.242 | **0.793** | 0.109 | **0.071** | **0.301** |
| AV-Cloud-sim-SH | 0.489 | 0.781 | 0.187 | 0.071 | 0.238 | 0.256 | 0.915 | 0.110 | 0.077 | 0.324 |

Table 4: Detailed Comparison with state-of-the-art methods on scenes of RWAVS validation set. For each metric on each scene, the top 1 value is highlighted in bold, the second best value is underlined.

In Table 4, we compare our results to state-of-the-art methods and provide detailed results across the four scenarios - Office, House, Apartment and Outdoor included in the RWAVS dataset [29]. We average the values across five acoustic metrics (explained as Section 4.1) for each scene. Overall, our method achieves higher accuracy across various scenarios, particularly in challenging environments like Apartment, House, and Outdoor, where the reference sound features distinctive reverberation effects and the scene exhibits complex geometry. Our approach shows significant advantages in metrics like perceptual quality (DPAM), reverberation effect (RTE), and binaural spatial effect (LRE), demonstrating its superior ability to capture acoustic properties and spatial effects through our point-based representation and point-based rendering pipeline.

## A.5 Spherical Harmonics Degree

Time filters are crucial for adjusting the energy distribution on the time domain for the output audio, adapting the reverberation effect, and rendering a more accurate spatial effect. By implementing the time filters with Spherical Harmonics parameters as outlined in Section 4, we are able to reduce the parameter count by over 50% while maintaining competitive accuracy, as shown in the experiments in Section 4.2. For our main experiments, the SH-based model variants *AV-Cloud-SH* and *AV-Cloud-sim-SH* use Spherical Harmonics degree 1.

In this section, we explore how varying the SH degree impacts model performance. We adjust the degree from 1 to 3 for the *AV-Cloud-sim-SH* model, and the results are shown in Table 5. Increasing the SH degree improves the accuracy of reverberation effect (RTE) and perceptual quality (DPAM), while with similar performance

on ENV and MAG. SH with degree 2 achieves the best LRE accuracy, which shows that stereo spatial effect accuracy does not necessarily improve with a higher SH degree.

| SH-Deg | # Params | MAG | LRE | ENV | RTE | DPAM |
|--------|----------|-------|-------|-------|-------|-------|
| SH-1 | **0.51M** | **0.359** | 0.996 | **0.147** | 0.076 | 0.291 |
| SH-2 | 0.84M | 0.361 | **0.983** | **0.147** | **0.072** | 0.292 |
| SH-3 | 1.30M | **0.359** | 0.993 | **0.147** | **0.072** | **0.282** |

Table 5: Ablation studies of Spherical Harmonics (SH) degree on RWAVS validation set.

## A.6 Audio-Visual Anchor Density Analysis

| #Anchors | MAG | LRE | ENV | RTE | DPAM |
|----------|-------|-------|-------|-------|-------|
| N = 64 | 0.354 | 0.978 | 0.146 | 0.078 | 0.280 |
| N = 128 | 0.354 | 0.963 | 0.146 | **0.074** | 0.281 |
| N = 256 | **0.351** | **0.936** | **0.145** | **0.074** | 0.276 |
| N = 512 | 0.354 | 0.958 | 0.146 | 0.075 | **0.275** |

Table 6: Robustness analysis of Audio-Visual Anchor density on RWAVS validation set.

To assess the impact of Audio-Visual Anchor density on system robustness, we conducted a density analysis experiment detailed in Table 6. We varied the K-Means cluster number (Section 3.2) from 64 to 25 to adjust the number of Audio-Visual Anchors. Most metrics remained stable, however, the spatial effect metric LRE improved by 4% when the anchor count increases from 64 to 256. Increasing the count to 512 slightly decreased accuracy of most metrics, except the perceptual quality DPAM, indicating that more anchors do not necessarily yield better results. Excessive anchors may complicate the re-weighting process and reduce accuracy.

## A.7 Visualization for Interpretation

To better understand how Audio-Visual Anchors contributes to the Visual-to-Audio Splatting process (as Section 3.3) in audio rendering, we visualize the attention weights of the Visual-to-Audio Splatting Transformer for some samples. As shown in Figure 6, we present three samples from different scenes in the RWAVS dataset. For each row, the sample includes a viewpoint image on the left and a plot of the SfM point clouds on the right. In the plot, the point clouds are shown in blue, and the Audio-Visual Anchor is visualized with a red point whose radius expands linearly with the attention weights: higher weights result in larger radii. The black triangle indicates the emitter, the green cross represents the listener, and the green divergent perspective starting from the listener shows the head orientation of the listener. The visualization typically shows anchors closer to the listener, key anchors along the boundary, and the anchor nearest to the emitter contributing more to the splatting process. This shows besides aware of listener's location in the 3D world coordinate system, our splatting transformer is able to localize the emitter and assign the anchor closest to it higher attention weights for spatial audio effect learning. Based on these capability, it is possible to extend our point-based scene representation to facilitate other 3D audio-visual cross-modal downstream tasks like scene understanding, object localization, tracking, generation for 3D scenes and more.

## A.8 Discussion: AVCS vs AV-Mapper

Both AV-Mapper, proposed in AVNeRF [29], and AVCS bridge the visual modality with spatial audio rendering. AV-Mapper relies on RGB and depth images, making it dependent on image data for viewpoint adaptation. In contrast, AVCS decouples the transfer function from images by using point-based (AV anchors) scene representation. Through anchor projection and the Visual-to-Audio Splatting Transformer, AVCS adapts to view poses using a view-independent 3D scene representation, converting monaural reference sound into stereo audio at the listener's viewpoint. Each query in AVCS corresponds to a frequency band, and the output is an integrated Relative Vector embedding the viewpoint's pose relative to the 3D anchor representation. AVCS involves explicit projection logic that adapts to the viewpoint pose in the 3D world.

The advantages of AVCS include: 1) Reusable anchor representation for all viewpoints, improving generalization to novel viewpoints with greater accuracy, fewer parameters, and faster inference as shown in Table 4.2. 2) Explicit adaptation to the 3D coordinate system, eliminating the need for visual rendering to obtain cues, resulting in more efficient and accurate audio-visual synchronization.

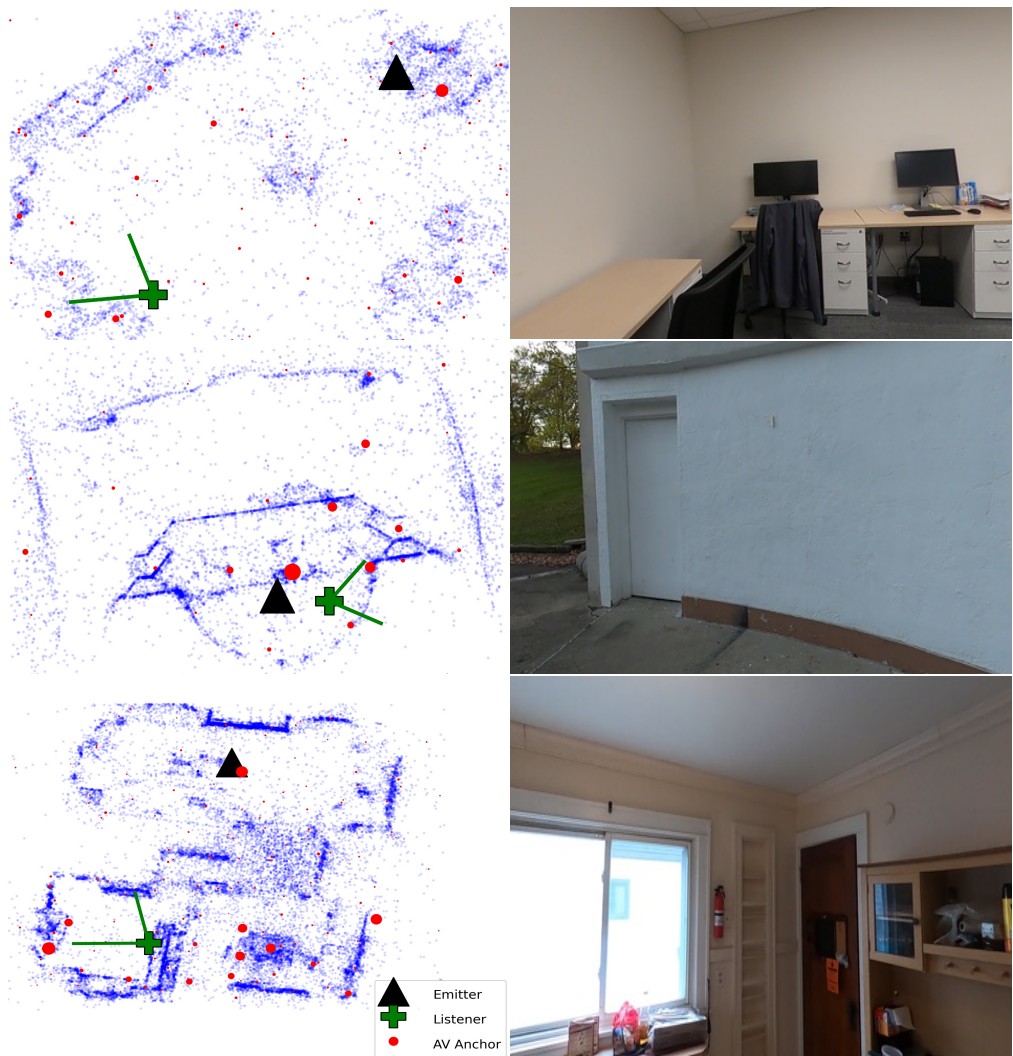

Figure 6: **Visualization for AV-Cloud Interpretation.** The black triangle indicates the emitter, the green cross represents the listener, and the green divergent perspective starting from the listener shows the listener's head orientation. Point clouds are shown in blue, and the Audio-Visual Anchor is visualized with a red point whose radius is proportional to attention weights. Our splatting transformer is able to localize the emitter and assign higher attention weights to the anchor closest to it for spatial audio effect learning.

## A.9 Limitations

Our system depends on camera calibration to obtain SfM points. In environments with limited visual data or inconsistent calibration spatial audio rendering quality may be of lower quality. In addition, noisy real-world audio captures could present a challenge. In such scenarios ambient noise can be present and reduce the fidelity of rendered audio. Our current system is designed for real-time processing and its components are thus of lightweight nature and are not designed to cope with extreme noisy cases. While our demo successfully managed sounds like wind and bird chirping, noisy environments remain a limitation especially when the reference sound is clean but the GT upon which the model was trained contains noise. Future solutions could include additional pre-processing or processing techniques such as noise reduction, sound separation, design of a higher-capacity model, or using generative models to compensate for reconstruction results.

## A.10 Broader Impacts

This work could contribute broadly to the next generation of spatial media applications such as virtual reality, gaming, and telepresence. In these applications, high-fidelity spatial audio could enhance immersion and realism,

and complement the visuals. While many broad aspects are possible for enhancement of the experience of spatial media, care should be taken to ensure that such immersive technologies are used responsibly and do not negatively affect the well-being of users. Also, there might be risks involved in using this methodology not for the intent it was designed, i.e. rendering fake outputs that may represent false interpretation of the scene or scam users. These risks are general risks and applicable to most typical generative rendering approaches.

