# OpenReview forum: "AV-Cloud: Spatial Audio Rendering Through Audio-Visual Cloud Splatting"
_NeurIPS.cc/2024/Conference — NeurIPS 2024 poster_

### Official Review · Reviewer_RM9S · 2024-07-13

**Soundness:** 3
**Presentation:** 4
**Contribution:** 3
**Rating:** 6
**Confidence:** 3

**Summary:**

The paper proposes AV-Cloud, a framework for high-quality spatial audio rendering in 3D scenes without relying on visual cues. AV-Cloud addresses issues in current audio-visual rendering methods, such as audio lag and dependence on visual rendering quality, by introducing Audio-Visual Anchors and the Audio-Visual Cloud Splatting module. These components facilitate the generation of viewpoint-specific spatial audio synchronized with visual content. The method demonstrates superior performance on multiple benchmarks, outperforming existing baselines in audio reconstruction accuracy, perceptual quality, and acoustic effects.

**Strengths:**

1. The concept of using Audio-Visual Anchors and Cloud Splatting to decouple audio rendering from visual rendering is interesting.
2. The paper demonstrates comprehensive experimentation and robust evaluation across multiple benchmarks.
3. The paper is well-structured and the presentation of the framework is clear. The figures and supplement examples help the readers better understand.
4. The proposed method addresses critical issues in real-time audio-visual rendering.

**Weaknesses:**

1. The mathematical formulation of the Audio-Visual Cloud Splatting module could be more detailed. For instance, Equation (2) introduces the softmax function applied to the relative vectors and visual features, but the reason behind this specific formulation and its implications are not sufficiently explained. Clarifying how the weights $a_{ki}$ are computed and how they influence the final output would enhance understanding.
2. The technical derivation of the Spatial Audio Render Head (SARH) lacks depth. Specifically, the process described in Equations (4) and (5), where the mixture mask $m_m$ and the difference mask $m_d$ are used to compute the left and right channel outputs, is not fully elaborated. The significance of these masks and their impact on the final audio quality are not clearly discussed. Additionally, the role and impact of the convolution modules within the residual structure (Figure 3) are not sufficiently explained.
3. While the method shows strong performance on benchmarks and some real-world examples, the provided examples are too idealized and lack challenging elements like interfering sound (e.g., crowd noise). I think the robustness of AV-Cloud in more complex and noisy real-world environments should also be validated.

**Questions:**

See Weaknesses.

**Limitations:**

The authors mention the limitations of their approach's challenges and potential drawbacks. The reliance on camera calibration and the potential issues with noise in real-world audio recordings are noted. Additional imitations can be found in the Weaknesses section

---

> ### Author Rebuttal · Authors · 2024-08-07
>
> We thank the reviewer for a thoughtful review, valuable feedback, and recognizing the innovative use of Audio-Visual Anchors and Cloud Splatting, comprehensive experimentation, and clear presentation. We address the questions and specify the intended revisions below.
>
> **W1: AVCS Explanation**
>
> In the general response section we provided a more detailed explanation of the AVCS module. We place the details relevant to the reviewer's raised point below for convenience as well.  We will add these details to the methods upon revision.
> The output of the Visual-to-Audio Splatting Transformer can be defined as follows:
>
> 1) **Attention Mask a_{ki}**: Indicates the contribution weight of each anchor, showing how much each anchor influences the spatial audio effect. Weighted sum the latent Audio Embedding e_i  to get **mixture Audio Embedding** e’. The mixture mask m_m is derived from e’ through an MLP.
>
> 2) **Output**: The final integrated Relative Vector embedding of anchors w.r.t the target viewpoint pose. The difference mask m_d is derived from the Output embedding through an MLP.
>
> In Equation (2), the softmax function is applied to compute attention weights a_{ki}. **This softmax function normalizes the weight of each anchor contribution**, enhancing the spatial audio effect to match the listener's perspective pose. Higher weights indicate greater influence, allowing the AVCS module to dynamically adjust the anchor contribution for audio rendering based on the listener's viewpoints. Please see the visualization examples in Figure 6.
>
>
> **W2: Spatial Audio Render Head (SARH)**
>
> SARH utilizes a one-layer residual structure with two convolution modules to enhance stereo output from Equation (4). The skip connection directly takes the output stereo channels from Equation (4) S_L and S_R. And the residual path includes:
>
> - **Time Filters**: Adjust the energy distribution of the mixture output spectrogram S_m across the time domain to match reverberation effects.
>
> - **Conv2D Layers**: Smooth and enhance the time-frequency domain energy distribution post Time Filters. The input to the Conv2D layers consists of four parts:
>     - Output of the Time Filters convolved with the mixture spectrogram S_m.
>     - Mixture mask m_m​.
>     - m_L=−m_d for the left channel or m_R=+m_d for the right channel post-process.
>     - Original stereo output from Equation (4): S_L for the left channel or S_R for the right channel post-process.
>
> The Conv2D layers process these inputs to generate the left and right residual channels. The final result is obtained by adding these residual outputs to the original S_L and S_R spectrograms from Equation (4).
>
> **Contribution of key components**
>
> We studied and verified the contribution of key components in SARH in ablation studies described in Sec 4.3 and Table 2. In particular:
>
> - **Baseline Model** (w/o AVCS and residual convolution modules): Utilize MLP to predict two acoustic masks. Compared to the full AVCloud, the baseline does not perform well across all five metrics, indicating limited generalization to novel views when relying solely on viewpoint poses. (Line 302-305)
>
> - **w/o AVCS**:
>      - Replace AVCS module with an MLP.  w/o AVCS is unable to capture the relationship between left-right channel energy and viewpoint poses well, reducing the LRE accuracy by 20% compared to our full AVCloud (Line 306-310).
>     - Compared with the Baseline, the **residual convolution module** can improve perceptual quality (0.357 -> 0.318), reverberation effect (0.13 -> 0.08), magnitude (0.488->0.468) and spatial effect metrics (1.329 -> 1.124) largely.
>
> - **w/o time**: Removing **Time Filters** to study its effect. Reverberation time error increases from 0.074 to 0.084, and LRE error increases by 5%, demonstrating the Time Filters’ crucial role in adjusting time-domain energy distribution, impacting reverberation time and overall acoustic quality. (Line 327-329)
>
> As suggested, we conducted two additional ablation experiments to study the effect of 1) two binaural masks and 2) Convolution 2D layers in the residual block.
>
> - **w/o 2 masks**: Direct prediction of two-channel spatial effect transfer mask from the AVCS output (instead of predicting difference mask and mixture mask separately). As a result, LRE significantly increases from 0.936 to 0.983.
>
> - **w/o conv2d**: Removal of 2D convolutional layers in the residual block and retaining Time Filters only. Without Conv2D layers, the Time Filter module handles reverberation but fails to distinguish between left and right channels well, indicated by LRE increasing from 0.936 to 1.019. Other metrics remain relatively stable. The need for Conv2D layers stems from reliance of residual blocks to post-process in the frequency-time domain.
>
> |Variants | MAG | LRE | ENV | RTE | DPAM|
> | -------- | ---- |---- |---- |---- |---- |
> |AVCloud (Full)|0.351|0.936|0.145|0.074|0.276|
> |w/o 2 masks|0.357|0.983|0.147|0.070|0.280|
> |w/o conv2d|0.359|1.019|0.148|0.074|0.280|
>
> We will include these results with detailed explanations and analysis in our final revision.
>
> **W3: Challenging elements such as interfering sounds**
>
> In our real demo, we successfully handled sounds like wind and bird chirping since the reference input captured these sounds. However, we indeed acknowledge that the scenario of dealing with highly noisy environments would pose a limitation to the approach, particularly scenarios where the ground truth contains noise, but the reference sound does not. Moreover, our current system is designed for real-time processing and its components are thus of lightweight nature and are not designed to cope with extreme noisy cases.
>
> Future solutions could be to consider additional pre-processing/processing such as noise reduction, sound separation, or the design of a higher capacity model. These are beyond our current scope but could be integrated with future developments. We appreciate the reviewer’s insight and will consider this in future works.

---

> > ### Comment · Reviewer_RM9S · 2024-08-12
> >
> > Thank you for addressing most of my concerns. I appreciate your effort on the additional ablations, and please make sure to include them in a revision. I’m happy to increase my score to WA.

---

> > > ### Author Response · Authors · 2024-08-13
> > >
> > > We want to thank the reviewers again for very helpful feedback and discussion. And we want to ensure the reviewers that we have their feedback and resulting revisions to algorithms description (W1, W2 to the Method Section), additional ablation studies analysis (W2 to the Ablations Section) and further limitation discussion (W3 to the Limitations Section). We will incorporate these content into a revised version with the additional page and more extensive supplementary information, which can highlight our motivation and contribution at a better clarity level.

---

### Official Review · Reviewer_ka62 · 2024-07-15

**Soundness:** 3
**Presentation:** 3
**Contribution:** 3
**Rating:** 7
**Confidence:** 1

**Summary:**

A novel approach for rendering high-quality spatial audio in 3D scenes, called AV-Cloud, is proposed. This method synchronizes with the visual stream without relying on or being explicitly conditioned by visual rendering, enabling immersive virtual tourism through real-time dynamic navigation of both audio and visual content. Unlike current audio-visual rendering methods that depend on visual cues and may suffer from visual artifacts causing audio inconsistencies, AV-Cloud overcomes these issues. It uses a set of sparse AV anchor points, forming an Audio-Visual Cloud derived from camera calibration, to represent the audio-visual scene. The Audio-Visual Cloud allows for the generation of spatial audio for any listener location. A novel module, Audio-Visual Cloud Splatting, decodes these AV anchor points into a spatial audio transfer function for the listener’s viewpoint, which is then applied by the Spatial Audio Render Head module to transform monaural input into viewpoint-specific spatial audio. This approach eliminates the need for pre-rendered images and efficiently aligns spatial audio with any visual viewpoint. The results are satisfying.

**Strengths:**

1. The AV anchors strategy seems to be interesting and effective for audio-visual scene representation. The Audio-Visual Cloud Splatting is novel for AV tasks but more likely to be a Q-former.
2. The experiment results are good and ablations are clear.

**Weaknesses:**

As I mentioned in the strengths, the Audio-Visual Cloud Splatting seems to be a Q-former like module.

**Questions:**

What is the difference between the AVCS and Q-former?

**Limitations:**

The authors adequately addressed the limitations.

---

> ### Author Rebuttal · Authors · 2024-08-07
>
> We thank the reviewer for a thoughtful review, valuable feedback and recognizing the novelty of AV Anchors for 3D audio-visual scene reconstruction.
>
> **W1 & Q1 Difference between AVCS and Q-former**
>
> While AVCS and Q-former are transformer-based structures, they serve different purposes and utilize transformer outputs in distinct ways.
>
> **Q-former** serves as an intermediary between a frozen image encoder and a frozen Language Model. Its query is a set of learnable vectors designed to extract visual features from the frozen image encoder. The query acts as an information bottleneck, providing the most useful visual features for the Language Model to generate the desired text output.
>
> In contrast, **AVCS** is designed to **learn features that adapt to view poses from a 3D scene representation to derive the audio spatial effect transfer function**. This function converts monaural reference sound into stereo audio at the listener’s viewpoint. Each query in AVCS corresponds to a specific frequency band, and the output is the integrated Relative Vector, embedding the viewpoint pose relative to the 3D anchors' scene representation. In contrast to Q-former, **AVCS involves explicit projection logic that adapts to the viewpoint pose in the 3D world, rather than implicitly extracting visual features relevant to audio**.
>
> Specifically, in the AVCS module, each Audio-Visual Anchor is projected to the head coordinate system of the target listener, and the anchor features are then integrated for each audio frequency band using the **Visual2Audio Splatting Transformer**. The attention mask indicates the contribution weight of each anchor, showing how much each anchor influences the spatial audio effect. This mechanism dynamically adjusts the contribution weights of anchors for audio rendering based on the listener's viewpoint. **The outputs and attention weights are used to derive the mixture and difference audio masks separately for the audio transfer function.**
>
> The Visual-to-Audio Splatting Transformer works as follows:
>
> **Input**
>
> 1) **Query**: Frequency embedding for each audio frequency band. Shape (F, C)
> 2) **Key**: Combined RGB visual feature of each anchor and its Relative Vector in the listener's head coordinate system. Shape (N, C)
> 3) **Value**: Relative Vector. Shape (N, C)
>
> **Output**
>
> 1) **Attention Mask a_{ki}**: Indicates the contribution weight of each anchor, showing how much each anchor influences the spatial audio effect. Shape (F, N). Weighted sum the latent Audio Embedding e_i (N, C) to get **mixture Audio Embedding** e’ (F, C)
> 2) **Output**: The final integrated Relative Vector embedding of anchors with respect to the target viewpoint pose. This embedding is highly relevant to the 3D pose of the listener. Shape (F, C)
>
> In Equation (2), the softmax function is applied to compute attention weights a_{ki}. This softmax function normalizes the weight of each anchor contribution, enhancing the spatial audio effect to match the listener's perspective pose. Higher weights indicate greater influence, allowing the AVCS module to dynamically adjust the anchor contribution for audio rendering based on the listener's viewpoints. Please see the visualization examples in Figure 6.

---

### Official Review · Reviewer_JmUt · 2024-07-18

**Soundness:** 2
**Presentation:** 1
**Contribution:** 2
**Rating:** 5
**Confidence:** 4

**Summary:**

The paper explores the problem of generating 3D audiovisual scenes – that is, generating 3D scenes with spatial audio. The proposed approach, AV Cloud, uses anchor points obtained from Structure-from-Motion (SfM) points. The anchors are then used with an AV Cloud splatting module which decodes the visuals and the audio. Experiments are done on RWAVS and Replay-NVAS with comparisons done with several prior works.

**Strengths:**

– 3d audiovisual scene generation is a really interesting problem to solve. WHile there is considerable literature on visual scene generation, generating 3d visual scene is an interesting problem with real-world applications.

– The model claims to be able to generate the audio and the visuals in parallel. Essentially unlike prior work it decouples the generation of two modalities by not using the generated visuals for generating the audio.

– On objective metrics, the paper claims to make good improvements

----
increased score after rebuttal

**Weaknesses:**

– The paper is a bit difficult to follow – especially the key part of AudioVisual anchor points.

– First, a short primer on SfM is desirable, even if it is in Appendix. More importantly though, it is not clear why it makes sense to use SfM points and clustering on top of them to model AV anchor points and generation of spatial points. Why does it make sense to use SfM points or anchors derived from them as the starting point for AV generation ? What relation the anchors have with audio which motivates the fact that these anchors can be used for audio generation ?

– Second, the details of AV anchor points are fuzzy. The visuals are used for SfM points which are then clustered to get the anchors. Where is the audio into picture here ? Are these anchors visual only ? If so, why are we calling it AV Anchors ?

– In prior works, for example AV-Nerf, there is an an explicit AV-Mapper which learns the audio visual relations through which the spatial audio generatio happens. Here Visual2Audio splatting transformer is expected to model that ?

– For the subjective tests, it would be good to actually get proper subjective ratings on the generated spatial audio. The current preference numbers are not very informative. Getting the spatial audio rated with respect to their quality and spatial characteristics would be much more meaningful.

– Since NAF, INRAS and other works are considered here - I think it would be good to reference NACF ([R1]) below. NACF specifically focuses on using visuals and is ideal for comparison.

[R1] Neural Acoustic Context Field: Rendering Realistic Room Impulse Response With Neural Fields

**Questions:**

Please address the questions below.

**Limitations:**

Yes.

---

> ### Author Rebuttal · Authors · 2024-08-07
>
> We thank the reviewer for a thoughtful review, valuable feedback and recognizing the importance of 3D audio visual scene synthesis and our contribution of proposing the novel parallel pipeline for audio and visual rendering. We address the questions and specify the intended revisions below.
>
> **W1 - Primer on SfM**
>
> SfM (Structure from Motion) reconstructs a 3D environment from a sequence of 2D images. SfM points are distinct and recognizable points in a scene detected in images. SfM points are used to determine the camera viewpoint and 3D coordinates leading to a detailed 3D point cloud. In Line 64-65, we included the reference for SfM [1] and SfM details are described in [1, Sec. 4.2]. As suggested by the reviewer, we will include a primer for SfM in the appendix that expands upon the short abstract of SfM and SfM points above and includes some key details from [1].
>
> **W2 - SfM and their clustering for AV generation**
>
> We found SfM points to be very informative to AV reconstruction due to:
> 1) They capture detailed 3D scene geometry, representing the physical boundaries and surfaces from which sounds can reflect, diffract, or be absorbed. This **geometric information** is key for rendering realistic audio-visual scenes and can be reused by any emitter and listener within the scene. Recent audio renderers [42, 32] also use point-based representations, showing more effective generalization with the same training data (Lines 109-128).
> 2) SfM points are also used to initialize visual renderers, e.g. 3D-GS[3], allowing for **parallel synchronization** of audio and visual rendering.
>
> **Clustering**: Clustering reduces the density of raw SfM point clouds, enhances computational efficiency while maintaining key geometric boundaries and surfaces (Lines 144-146, visualized in Figure 6).
>
> We will further clarify our motivation in the introduction and method sections.
>
>
> **W3 - AV anchors their use for AV generation**
>
> After clustering, we initialize the anchor features using: 1) **3D coordinates**; 2) **RGB values**; and 3) **Latent Audio Embedding** e_i​  (Lines 140-151)  for a point-based audio-visual representation. Specifically, the Audio Embedding captures how the anchor region contributes to sound propagation for different listener viewpoints, similar to boundary bounce points in INRAS [42].
>
> AV Anchors derive the spatial audio transfer function via the AVCS module (Sec 3.2).  We have included an additional, more practical, and detailed explanation of the AVCS module in our response to all reviewers and we will incorporate it in the revision as well. Each anchor is projected into the listener's head coordinate system, adapting to head orientation with the following components:
> 1) **3D Coordinates**: Calculate the Relative Vector, serving as Key and Value for the Visual2Audio Splatting Transformer.
> 2) **RGB Values**: Positional encoding for the Key to enhance scene understanding and distinguish between Anchors at different locations (lines 170-174).
> 3) **Latent Audio Embedding**: Weighted by the Attention Mask to get the mixture Audio Embedding. The mixture mask is derived from this embedding through an MLP, manipulating mixture sound magnitude changes.
> The final integrated Relative Vector from the transformer is used to obtain the difference mask through MLP.
>
> **W4 - Difference between AVCS and AV-Mapper**
>
> AV-Mapper (AVNeRF) relies on RGB and depth images, making it dependent on image data to adapt to viewpoint changes. In contrast, AVCS module decouples the transfer function from images, using point-based (AV anchors) scene representation. By anchor projection and the Visual-to-Audio Splatting Transformer, our approach adapts to view poses from a **view-independent 3D scene representation**.
>
> **Advantages**:
> 1) **Reusable anchor representation for all viewpoints**, enabling better generalization to novel viewpoints with better accuracy, fewer parameters, and higher inference speed (Table 1).
> 2) **Explicit adaptation to 3D world coordinate system**, eliminating the need for visual rendering to obtain strong visual cues. The approach can achieve more efficient and accurate audio-visual synchronization without reliance on visual render results to obtain strong visual cues.
>
> **W5 - Subjective Tests Justification**
>
> We particularly set the subjective test instructions to target evaluation of the alignment of spatial audio effects with visual content. Participants were instructed to select the video where spatial audio matches the visual content, focusing on left-right ear effects and overall synchronization. Instructions included:
>
> “… Select the video that features the spatial audio effect best matching the visual content.… Pay attention to the varying left-right ear spatial effects… Evaluate each video comprehensively for spatial effects, audio continuity, and quality…”
>
> This test complements quantitative metrics like LRE, providing a comprehensive assessment. Our method received 48% of the votes, outperforming other methods, demonstrating its effectiveness in creating an immersive audio-visual experience.
>
>
> **W6 - Comparison with NACF**
>
> We thank the reviewer for recommending to include NACF in the evaluations. Similar to INRAS [32] and NAF [42], NACF does not include viewpoint-based reweighting, which leads to less effective representation. AVCloud, with dynamically reweighted Audio-Visual Anchors, significantly improves audio rendering metrics. The Spatial Render Head also enhances reverberation effects, achieving a more accurate acoustic metric (RTE). We will include these references and results in our revision.
>
> RWAVS Dataset:
> |Method | MAG | LRE | ENV | RTE | DPAM|
> | -------- | ---- |---- |---- |---- |---- |
> |NACF|0.459|1.364|0.176|0.138|0.506|
> |AVCloud (Ours)|0.351|0.936 |0.145|0.074|0.276|
>
> Replay-NVAS Dataset:
> |Method | MAG | LRE | ENV | RTE | DPAM|
> | -------- | ---- |---- |---- |---- |---- |
> |NACF|0.298|0.722|0.079|0.332|0.544|
> |AVCloud (Ours)|0.180|0.600|0.052|0.065|0.234|

---

> > ### Comment · Reviewer_JmUt · 2024-08-11
> > **After rebuttal**
> >
> > Thanks for the detailed response. The rebuttal add in a lot of clarity. I think the paper will need a good amount of change to incorporate all the clarity and additional results. I have increased the score.

---

> > > ### Author Response · Authors · 2024-08-13
> > >
> > > We want to thank the reviewers again for very helpful feedback and discussion. And we want to ensure the reviewers that we have their feedback and resulting revisions to motivation explanation (W2 to the Introduction Section), algorithms description (W2, W3 to the Method Section), comparison discussion with addition results (W4, W5, W6 to the Experiment analysis) and primer content (W1 to the Supplementary Section). We will incorporate these content into a revised version with the additional page and more extensive supplementary information, which can highlight our motivation and contribution at a better clarity level.

---

### Author Rebuttal · Authors · 2024-08-07

We thank the reviewers for their thoughtful reviews and valuable feedback. In this general section, we wanted to provide a more **detailed explanation of the Audio-Visual Cloud Splatting (AVCS) module**, as several reviewers have suggested.

The AVCS module is one of the key contributions of our work. It receives as input point-based audio-visual scene representation, AV Anchors, and decodes them into spatial audio transfer function (Sec 3.2). The spatial audio transfer function is of the form of two acoustic masks which convert the monaural reference sound into stereo audio.

In the AVCS module, each Audio-Visual Anchor is projected to the head coordinate system of the target listener. The anchor features are then integrated for each audio frequency band using the **Visual2Audio Splatting Transformer** which works as follows:

**Input**:
1) **Query**: Frequency embedding for each audio frequency band. Shape (F, C)
2) **Key**: Combined RGB visual feature of each anchor and its Relative Vector in the listener's head coordinate system. Shape (N, C)
3) **Value**: Relative Vector in the listener's head coordinate system. Shape (N, C)

**Output**:
1) **Attention Mask a_{ki}**: Indicates the contribution weight of each anchor, showing how much each anchor influences the spatial audio effect w.r.t each frequency band. Shape (F, N). Weighted sum the latent Audio Embedding e_i (N, C) using attention masks to get mixture Audio Embedding e’ (F, C).
2) **Output**: The final **integrated Relative Vector** embedding of anchors for each frequency band w.r.t the target viewpoint pose. This embedding is highly relevant to the 3D pose of the listener. Shape (F, C)

In Equation (2), the softmax function is applied to compute attention weights a_{ki}. This softmax function normalizes the weight of each anchor contribution, enhancing the spatial audio effect to match the listener's perspective pose. Higher weights indicate greater influence, allowing the AVCS module to dynamically adjust the anchor contribution for audio rendering based on the listener's viewpoints. Please see the visualization examples in Figure 6.

The spatial audio transfer function consists of two masks: a mixture mask m_m​  and a difference mask m_d. The two masks then convert monaural reference sound into stereo audio at the listener’s viewpoint. We obtained the two masks from the AVCS module as follows:

1) **The mixture mask (m_m​)** is derived from the weighted sum of latent Audio Embeddings through MLP, showing mixture sound magnitude variation across viewpoint locations.
2) **The difference mask (m_d)**, derived from the final integrated Relative Vector (Output of Visual2Audio Splatting Transformer) through MLP, captures left-right channel differences relevant to the listener's 3D pose.

We will add the above explanations to our revision.

---

### Decision · Program_Chairs · 2024-09-25

**Decision:**

Accept (poster)

**Comment:**

This paper has received unanimous recommendation of acceptance. The recommendations after the rebuttal are more in favour of accepting this paper. Please make sure that the clarifications in the rebuttals are properly integrated into the final paper.